# Spectral PINNs: Fast Uncertainty Propagation with Physics-Informed Neural Networks

**Björn Lütjens** *
Human Systems Laboratory
MIT

**Catherine H. Crawford**
IBM Research

**Mark Veillette**
MIT Lincoln Laboratory

**Dava Newman**
Human Systems Laboratory
MIT & MIT Media Lab

## Abstract

Physics-informed neural networks (PINNs) promise to significantly speed up partial differential equation (PDE) solvers. However, most PINNs can only solve deterministic PDEs. Here, we consider *stochastic* PDEs that contain partially unknown parameters. We aim to quickly quantify the impact of uncertain parameters onto the solution of a PDE - that is - we want to perform fast uncertainty propagation. Classical uncertainty propagation methods such as Monte Carlo sampling, stochastic Galerkin, collocation, or discrete projection methods become computationally too expensive with an increasing number of stochastic parameters. For example, the well-known spectral or polynomial chaos expansions achieve to separate the spatiotemporal and probabilistic domains and offer theoretical guarantees and fast computation of stochastic summaries (e.g., mean), but can be computationally expensive to form. Our *Spectral PINNs* approximate the underlying spectral coefficients with a neural network and reduce the computational cost of the spectral expansion while maintaining guarantees. We derive the method for partial differential equations, discuss runtime, demonstrate initial results on the convection-diffusion equation, and provide steps towards convergence guarantees.

## 1 Introduction

Solving partial differential equations (PDEs) in fluid dynamics, astrophysics, or climate science is computationally expensive [1]. For example, solving a climate model at 1km resolution can take 10 days on a 5000-GPU supercomputer [2]. Decades of research in numerical methods [3] and physical approximations, such as parametrizations or closures [4], has achieved drastic reductions in the complexity of solving PDEs. However, numerical methods remain computationally expensive for high-dimensional problems or uncertainty quantification [5] and physical approximations require extensive domain knowledge and often sacrifice accuracy [4].

Recently, researchers in the field of physics-informed neural networks (PINNs) have proposed to combine numerical methods with deep neural networks (NNs) [6, 5]. The combination builds a perfect team: NNs have fast inference times despite approximating high-dimensional functions, but lack guarantees and constraints [7]. Numerical methods are slow for high-dimensional problems, but have rigorous convergence guarantees and physical constraints [8]. Each balancing the other's weaknesses, physical knowledge has been integrated into NNs as inputs [9, 10, 11], loss function [6], latent

---

*Corresponding author: `lutjens@mit.edu`

35th Conference on Neural Information Processing Systems (NeurIPS 2021), Sydney, Australia.

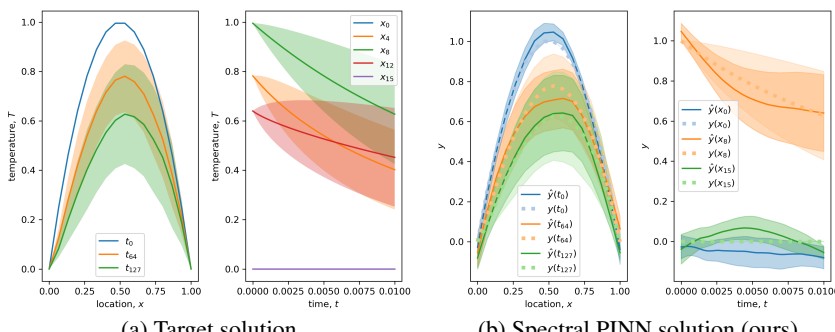

|   (a) Target solution   |   (b) Spectral PINN solution (ours)   |

Figure 1: **Spectral PINNs.** Initial results show that the spectral PINNs in Fig. 1b-solid can approximately match the mean (line) and standard deviation (shade) of the target solution in Fig. 1a and Fig. 1b-light-shade-dotted (with $y=u$). Importantly the approximated standard deviation also captures the growing trend towards the center location ($x = 0.5$) and growing time, $t = 1$ (far-right-plot, orange).

state [12, 13, 14, 15], analytic constraints [16, 17], or evaluation function [18, 19]. From the other side, NNs have been embedded into numerical methods by approximating the PDE's parameters [20, 6], dynamics [21, 12], residual [22, 23], differential operator [24, 25], or solution [6, 26]. Our *Spectral PINNs* leverage NNs to approximate the solution of a PDE, creating a learning-based "surrogate model" [27, 12] that can be used for fast propagation of uncertainties or sensitivity analyses.

PINNs have achieved to speed-up PDE solvers by 1-3 orders of magnitude [28, 26], but are mostly limited to deterministic PDEs [5]. In practice, many parameters of PDEs are partially unknown and uncertainty quantification plays a vital role in the development of real-life safety-critical systems [29], climate risk estimation [30], or parameter inference [27]. A small set of PINNs has targeted stochastic differential equations [31, 32, 33], but has been developed rather independently from the existing tools in uncertainty quantification (UQ) [27]. Our work builds a bridge between UQ and deep learning by combining well-known spectral expansions with neural networks.

Uncertainty propagation, the UQ subfield of our interest, aims to calculate statistical summaries of the solution given knowledge about parameter distributions [27]. We can roughly divide methods for uncertainty propagation in intrusive and non-intrusive techniques. Intrusive techniques, such as Stochastic Galerkin [34], require domain knowledge of the PDEs and modifiability of the PDE solver which often does not exist [8]. Non-intrusive techniques, such as Monte Carlo sampling [35], collocation methods [36], or discrete projection [36], are purely data-driven methods that only assume solver-generated datasets. The runtime complexity of many non-intrusive techniques, however, can scale cubically with the number of stochastic parameters [34]. Purely deep learning-based generative models such as GANs [37] or VAEs [38] can offer fast uncertainty propagation via Monte Carlo sampling, but often come without theoretical guarantees [39, 40]. Our method is non-intrusive, faster than conventional UQ methods, and leverages a widely adopted UQ technique called spectral expansion or polynomial chaos expansion (PCE) offering some guarantees [41, 42, 27].

In particular, we propose to approximate the spectral coefficients with NNs. The integration of NNs into spectral expansions enables fast calculation of statistical summaries, sensitivity analyses [43], and the potential for theoretical guarantees of optimality or convergence [27]. The closest to our work proposes NNs to estimate the coefficients of an arbitrary polynomial chaos [44], but has only been demonstrated on a 1D stochastic elliptic diffusion equation, not on PDEs, and is intrusive [45].

We contribute *Spectral PINNs*, specifically:

- The extension of spectral PINNs to PDEs, spatiotemporal, parabolic, hyperpolic, and evolutionary equations.

- A scalable, mesh-free, and data-driven method for non-intrusive uncertainty propagation in potentially high-dimensional parameter and spatiotemporal spaces.

- A potentially faster method for uncertainty propagation than traditional UQ methods, such as stochastic Galerkin or collocation.

- The first demonstration of *Spectral PINNs* on the convection-diffusion equation.

# 2 Approach

Quantifying uncertainties in solutions to stochastic PDEs (SPDEs), via computing large ensembles is computationally expensive. In this section, we approximate the solution using a spectral expansion, and estimate the coefficients of that spectral expansion with NNs to build a fast and accurate surrogate for uncertainty propagation.

## 2.1 Problem definition

First, we define a stochastic PDE with the solution, $u$, and stochastic parameters, $\boldsymbol{Q}$, as:

$$\mathcal{N}_{t,x}[u(t,x;\omega); \boldsymbol{Q}(x;\omega)] = 0, \ \forall \ t \in D_t, \ x \in D_x, \ \omega \in \Omega,$$
$$\mathcal{B}_{t,x}[u(t,x;\omega)] = 0, \ \forall \ t \in \partial D_t, \ x \in \partial D_x,$$

with spatial domain, $D_x \subset \mathbb{R}^{n_x}$, temporal domain, $D_t \subset \mathbb{R}_+^{n_t}$, sample space, $\Omega$, domain boundary, $\partial$, potentially nonlinear operator, $\mathcal{N}_{t,x}$, and initial and Dirichlet boundary conditions, $\mathcal{B}_{t,x}$.

We aim to learn a surrogate model, i.e, an approximation to the solution, $\hat{u}$:

$$\hat{u} : D_t \times D_x \times \boldsymbol{Q} \mapsto \mathbb{R}^{n_u}. \tag{1}$$

The domain of $\hat{u}$ are points of the form $(t, x, \boldsymbol{Q}(\xi))$, where $\boldsymbol{Q}(\xi)$ is a sample from a finite dimensional distribution of $\boldsymbol{Q}$, i.e. $\boldsymbol{Q}(\xi) = [Q(x_0; \xi), \dots, Q(x_{n_x-1}; \xi)]$ for some $n_x > 0$ and $x_0, \dots, x_{n_x-1} \in D_x$ and the realizations of a new stochastic basis, $\xi$. The solution then maps a distribution of parameters to a distribution of solutions.

## 2.2 The Local Convection-Diffusion Equation

We use the convection-diffusion equation to illustrate our method. This is a well-known evolutionary PDE that combines parabolic and hyperbolic equations and is conceptually similar to incompressible Navier-Stokes. The equation models the temperature distribution in a vertical ocean column over time as,

$$f = \frac{\delta T(t,x;\omega)}{\delta t} + w \frac{\delta T(t,x;\omega)}{\delta x} - \frac{\delta}{\delta x} \left( \kappa(x;\omega) \frac{\delta T(t,x;\omega)}{\delta x} \right), \tag{2}$$

with height, $x \in D_x = [0,1]$, time, $t \in D_t = [0, 0.01]$, source, $f = 0$, noise, $\omega \in \Omega$, univariate $(n_u{=}1)$ temperature, $u = T : D_t \times D_x \times \boldsymbol{Q} \mapsto \mathbb{R}^1$, stochastic diffusivity, $Q(x;\omega) = \kappa(x;\omega)$ (detailed in Appendix A.2), partial derivative, $\delta$, constant vertical velocity, $w = 100$, initial temperature, $T(0, x; \omega) = -4(x - 0.5)^2 + 1$, and Dirichlet boundary conditions, $T(t, 0; \omega) = T(t, 1; \omega) = 0$.

## 2.3 Spectral expansion

A core idea in our paper is to approximate the solution, $u$, as a spectral expansion, also called polynomial chaos expansion [42, 27]. The spectral expansion projects the solution onto a space of stochastic orthogonal polynomials. The solution then becomes a linear combination of spatiotemporal coefficients and stochastic polynomials. The projection is optimal in the L2 sense [27]. The spectral expansion approximates the solution as,

$$\hat{u}(t, x; \boldsymbol{\xi}) = \sum_{j=0}^{|A|-1} \hat{C}_{\boldsymbol{\alpha}_j}(t, x) \Psi_{\boldsymbol{\alpha}_j}(\boldsymbol{\xi}), \tag{3}$$

where $\hat{C}_{\boldsymbol{\alpha}_j}(t, x)$ are deterministic spectral coefficients of the expansion of the random variable $\hat{u}(t, x; \boldsymbol{\xi})$. In our approach we replace these coefficients with neural network function approximations that will be learned from data. The unknown stochastic domain $\omega \in \Omega$ is approximated by a transformed domain that can easily be sampled from, for example, a set of iid. Gaussians with $\boldsymbol{\xi} = \{\xi_0, \dots, \xi_{n_\xi-1}\} \sim \mathcal{N}(0, 1)^{n_\xi-1}$ with the truncated stochastic dimension $n_\xi \leq n_x$. We choose the set of polynomials, $\Psi_A(\boldsymbol{\xi})$, to transform a Gaussian distribution, specifically a multivariate orthogonal Gaussian-Hermite polynomials (as displayed and displayed and detailed in Appendix A.3.1). The polynomials and spectral coefficients are indexed by the multi-indices, $\boldsymbol{\alpha}_j \in A$, which contain index-pairs into the stochastic and polynomial degree, as detailed in Appendix A.3.2.

## 2.4 Spectral PINNs

The NN jointly estimates all $|A|$ spectral coefficients based on 2D spatiotemporal input pairs, $\{t, x\}$,

$$NN_{C_A}(t, x) := \hat{C}_A(t, x) : D_t \times D_x \to \mathbb{R}^{|A|}. \tag{4}$$

Using single input pairs makes our surrogate model mesh-free as the NN can be queried at any location to interpolate the training samples or collocation points. The NN is trained to approximate the spectral coefficients while only using a limited number of measurements as target data. We formulate the mean-squared error (MSE) loss for episode, $e$, and batch size, $B$, as:

$$L_e(t_b, x_b) = \frac{1}{B} \sum_{b=0}^{B-1} ||u(t_b, x_b; \boldsymbol{\xi}_b) - \sum_{j=0}^{|A|-1} \hat{C}_{\boldsymbol{\alpha}_j}(t_b, x_b) \Psi_{\boldsymbol{\alpha}_j}(\boldsymbol{\xi}_b)||_2^2, \tag{5}$$

where the realizations of Gaussian random vectors, $\boldsymbol{\xi}_b \sim \mathcal{N}(0, 1)^{n_\xi - 1}$ are shared between the target and approximated solution. The batch size is chosen to fit one solution sample, $B = n_t n_x$ with t-grid size, $n_t$, and x-grid size, $n_x$. The training and validation/test set are using different realizations of the random vector and the same $\{t, x\}$-pairs as grid points.

# 3 Preliminary Results

Figure 1 shows that the *Spectral PINNs* in Fig. 1b can successfully approximate the mean and standard deviation of the target solution Fig. 1a. Importantly the explicit formulation as spectral expansion allows us to compute the mean and standard deviation without any sampling as a function of the spectral coefficients, e.g., $\mu_u(t, x) = C_{[0,0,0]}(t, u)$ [27]. We can note that the *Spectral PINNs*-approximated standard deviation captures the growing trend towards the center location ($x = 0.5$) and increasing time ($t = 1$). While the accuracy of estimating the mean and std. dev. could be improved towards regions of high uncertainty ($x = 0.5, t = 0.05$) the plots indicate promise and we expect hyperparameter tuning to further increase the accuracy. Figure 4 shows the approximated spectral coefficients.

We used a 6-layer 2048-unit fully-connected neural network with ResNet blocks, in-/output normalization, and ReLu activation. The network was trained with the ADAM optimizer with learning rate, $lr = 0.001$, and $\beta = [0.9, 0.999]$ for $E = 8$ epochs in $\approx$1hr on one 5-year old Intel $i7-7500U$@2.70GHz quad-core CPU. The target data was generated with $n_t = 128$ temporal and $n_z = 16$ grid points and $\{n_{s,\text{train}} = 800, n_{s,\text{val}} = 200\}$ samples of the solved PDE. The maximum polynomial degree was chosen to be, $n_\xi = 3$, s.t. the number of spectral coefficients is $|A| = 10$.

## 3.1 Linear vs. cubic runtime complexity

Introducing NNs into uncertainty propagation (UP) methods aims to reduce runtime complexity while maintaining accuracy in conducting statistic analysis of high-dimensional PDEs.

**Runtime complexity.** Classical UP methods can scale cubically with the number of spectral coefficients in the worst-case [27]. For example, stochastic Galerkin requires to solve one system of $J(K + 1)$ equations [34] and collocation methods require to solve $M$ systems of size $J$ [36]. Here, $J$ is the number of spectral coefficients, $K$ the number of stochastic polynomials, $M$ the number of training samples and in our case $J=K=|A|=10$, $M=n_{s,\text{train}}=800$. In comparison, inference of our FCNN scales linearly with the number of outputs/spectral coefficients, $J$, assuming the network architecture is independent of $J$ [46]. In practice, the architecture depends on the number of outputs and future work will investigate this relationship. Further, a full comparison will include a discussion on how training time scales with the number of outputs and experimental plots.

**Convergence rate.** The final accuracy of collocation methods scales linearly in the number of samples, $M$ [36]. However, the convergence rate towards an accurate solution has poor scaling with the number of stochastic parameters $n_\xi$ as $O(M^{-\alpha_{\text{regularity}}/n_\xi})$. In comparison, MC sampling converges with $O(M^{-1/2})$ which is faster for a high number of stochastic parameters, $n_\xi$ [27]. Future work, will include a discussion on how the accuracy of NNs converges with the number of stochastic training samples, $M$, and stochastic dimension, $n_\xi$ [47], and provide examples with increased $n_\xi > n_{\text{KL}} = 3$ (see Appendix A.2 for current stochastic dimension).

**Memory complexity.** In some cases, leveraging NN-based surrogate models can not only reduce computational complexity but also memory complexity [23]. Our network, however, contains $n_w = 2n_{units} + n_{layers}n_{units}^2 + n_{units}|A| = 25M$ weights which occupy as floats $n_{weights}4B \approx 96MB$. As this is quite large, future work, will investigate pruning [48] and mixed-precision [49] methods.

## 4 Discussion and Future works

We contributed *Spectral PINNs*, a spectral method for fast uncertainty propagation of high-dimensional stochastic PDEs. We demonstrated the first application of NN-based spectral expansions on the convection-diffusion equation. There exist various avenues for future work. First, we would like to extend hyperparameter search to increase accuracy and decrease memory consumption. Second, our work assumes that the stochastic parameters are mutually independent and Gaussian which can be alleviated by an extension to arbitrary polynomial chaos expansion for arbitrary distributions [50]. Third, future work will investigate efficient sampling strategies to reduce the amount of training samples, such as Latin hypercube sampling [51] or Quasi Monte Carlo sampling [52]. Fourth, our work relies purely on FCNNs, but the approximation of as spatiotemporal coefficients naturally lends itself to RNNs [53], CNNs [54], GNNs [55], Transformers [56], or more to exploit spatiotemporal features and propagate uncertaintes of higher-dimensional systems. Lastly, we will continue the analysis of convergence guarantees based on recent work in converging PINNs [47].

## Acknowledgments and Disclosure of Funding

The authors greatly appreciate the discussions with Chris Hill, Shu-Yu Lin, Nicholas Mehrle, Yanni Yuval, Paul O'Gorman, Jaume Peraire, and Youssef Marzouk.

Research was sponsored by the United States Air Force Research Laboratory and the United States Air Force Artificial Intelligence Accelerator and was accomplished under Cooperative Agreement Number FA8750-19-2-1000. The views and conclusions contained in this document are those of the authors and should not be interpreted as representing the official policies, either expressed or implied, of the United States Air Force or the U.S. Government. The U.S. Government is authorized to reproduce and distribute reprints for Government purposes notwithstanding any copyright notation herein.

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

# A Appendix

## A.1 The local convection-diffusion equation

We generated $n_s = 1000$ samples of the stochastic convection-diffusion equation via a 2nd-order finite difference scheme in Python. We used the number of discretization points, $n_t = 128$ and $n_x = 16$. We used a 2nd order forward/central/backward difference scheme for the left boundary/interior/right boundary of the domain, respectively. The initial condition was then time-stepped with $T(t + 1) = T(t) + \Delta t \frac{dT(t)}{dt}$.

## A.2 Sampling the Stochastic Parameter during Training

We assume that the distribution over the diffusivity is known, for example, through domain knowledge, data assimilation, or Bayesian parameter estimation. Specifically, the diffusivity is assumed to follow an exponential Gaussian process (GP) with $\kappa(x; \omega) = \exp(Y_\kappa(x; \omega))$. The GP, $Y_\kappa(x; \omega)$, is defined by mean, $\mu_{Y_\kappa} = 3$, correlation length, $L = 0.3$, variance, $\sigma_{Y_\kappa} = 1.0$, exponent, $p_{\text{GP}} = 1.0$, and a covariance kernel that is similar to the non-smooth Ornstein-Uhlenbeck kernel:

$$\text{Cov}_{Y_\kappa}(x_i, x_j) = \sigma_{Y_\kappa}^2 \exp\left[-\frac{1}{p_{\text{GP}}}\left(\frac{|x_i - x_j|}{L}\right)^p\right]. \tag{6}$$

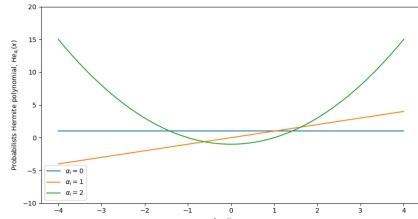

Figure 3: **Hermite Polynomials.** The basis functions for the first three Hermite polynomials, $\text{He}_i(x)$ for $i = \{0, 1, 2\}$. As the

During training we sample the GP via a Karhunen-Loève (KL) expansion [27]:

$$
\begin{aligned}
\kappa(x_i; \omega) &= \exp(Y_\kappa(x_i; \omega)) \\
&\approx \exp(\mu_{Y_\kappa} + \sum_{j=0}^{n_{\text{KL}}-1} \sqrt{\lambda_j}\phi_j(x_i)\xi_j).
\end{aligned}
\tag{7}
$$

Here, $\lambda_j$ and $\phi_j(x_i)$ are the $j$-th eigenvalues/-vectors of the covariance matrix respectively, s.t., $\sum_i^{n_x} \text{Cov}(x_j, x_i)\phi_j(x_i) = \lambda_j\phi_j(x_j)$. The KL-expansion has a maximum of $n_{\text{KL,max}} = n_x$ modes which have been truncated at $n_{\text{KL}} = 3$ modes. The GP which is defined by random variable, $\omega$, is then approximated by drawing iid. Gaussian random samples, $\boldsymbol{\xi}_b = \{\xi_0, ..., \xi_{n_{\text{KL}}-1}\}_b \sim \mathcal{N}(0,1)^{n_{\text{KL}}-1}$, and projecting them onto the eigenspace of the covariance matrix.

### A.3 Polynomial basis

#### A.3.1 Hermite polynomials

We choose the set of polynomials, $\Psi_A(\boldsymbol{\xi})$, to be a set of multivariate orthogonal Gaussian-Hermite polynomials, as displayed in Fig. 3:

$$
\begin{aligned}
\Psi_{\boldsymbol{\alpha}_j}(\xi_0, ..., \xi_{n-1}) &= \Pi_{i=0}^{n-1}\psi_{\alpha_{ji}}(\xi_i), \\
&= \Pi_{i=0}^{n-1}\text{He}_{\alpha_{ji}}(\xi_i), \\
\text{with } \xi_i &\sim \mathcal{N}(0,1),
\end{aligned}
\tag{8}
$$

with the one-term (monic) polynomials, $\psi_{\alpha_{ji}}$ of polynomial degree, $\alpha_{ji}$, and maximum degree, $n$. We are choosing the random vector of each stochastic dimension, $i \in \{0, ..., n-1\}$, to be a Gaussian, $\xi_i \sim \mathcal{N}(0,1)$ and use the associated probabilists' Hermite polynomials, $\text{He}_{\alpha_{ji}}$.

We chose Hermite polynomials, because we assumed the stochastic parameter, $\kappa$, to follow a Gaussian Process and Hermite polynomials are orthogonal with respect to a Gaussian measure [8]. For other distributions we choose other polynomials, e.g., uniform→Legendre, $\gamma$ →Laguerre, as listed in table 2.4 of [8]. Future work, will analyse which space of functions can be approximated by *Spectral PINNs* that rely on Gaussian-Hermite polynomials.

#### A.3.2 Multi-indices

We choose the vector of polynomial degrees, $\boldsymbol{\alpha}_j$, that builds the polynomial set, $\Psi_{\alpha_j}$ to be all combinations of degrees that sum to a maximum of $n - 1$. In other words, the multi-index, $\boldsymbol{\alpha}_j \in A$, with $j \in \{0, ..., |A|-1\}$ is defined by the total-degree multi-index set, $A = \{\boldsymbol{\alpha}_j \in \mathbb{N}_0^n : ||\boldsymbol{\alpha}_j||_1 = \sum_{i=0}^{n-1}\alpha_{ji} \leq n-1\}$ [27]. For example, $A = \{\boldsymbol{\alpha}_0, ..., \boldsymbol{\alpha}_{|A|-1}\} = \{[0,0,0], [0,0,1], [0,1,0], [1,0,0], [0,0,2], [0,1,1], [0,2,0], [1,0,1], [1,1,0], [2,0,0]\}$ and $|A| = 10$ for $n = 3$.

### A.4 Learned spectral coefficients

Figure 4 shows the approximated spectral coefficients.

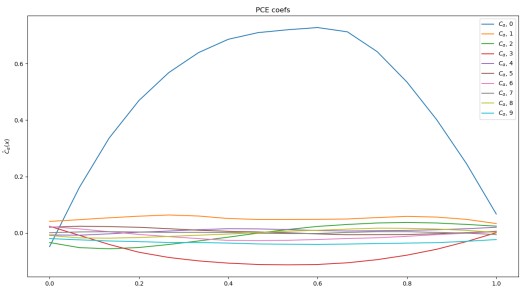

Figure 4: **Learned spectral coefficients.** The neural network outputs the first $|A| = 10$ spectral coefficients. We can observe that the first coefficient, $C_{[0,0,0]}$ (blue), approximates the mean and is of significantly larger scale than the other coefficients.

