# OpenReview forum: "Spectral PINNs: Fast Uncertainty Propagation with Physics-Informed Neural Networks"
_NeurIPS.cc/2021/Workshop/DLDE — DLDE Workshop -- NeurIPS 2021 Poster_

### Official Review · Reviewer_ZLzv · 2021-10-11
**Excellent paper with interesting idea**

**Confidence:** 3

**Review:**

The paper presents an extension of PINN to stochastic PDEs which contain partially unknown parameters. The key idea is to use "polynomial chaos expansions" to approximate the solution, thereby the name "Spectral PINNs" came from. I found the paper well-organized and clearly conveyed what the authors try to express. The preliminary results are also convincing enough to support their arguments. Ideally in a full-length paper, I would love to know more about the choice of the expansion, whether the choice is empirical or have any theories backbone. Also, more numerical shreds of evidence shall be desired since the current one only shows a comparison of mean and std.

**Score:**

3: Good paper

---

### Official Review · Reviewer_Tc1W · 2021-10-11
**An implementation of spectral expansion using NNs to solve stochastic PDEs**

**Confidence:** 3

**Review:**

The authors propose to use NNs to approximate the spectral coefficients of a spectral expansion used to approximate the solutions of a stochastic PDE. To this end, they propose Spectral Physics Informed Neural Networks (PINNs). Although the authors didn't discuss the choice of the approximation in detail, the paper is well written and suitable for the workshop.

**Score:**

3: Good paper

---

### Official Review · Reviewer_HHyy · 2021-10-11
**Nice direction of research requiring further investigation**

**Confidence:** 3

**Review:**

Neural networks are combined in this contribution with the polynomial chaos expansion technique to solve stochastic PDEs in order to efficiently quantifying uncertainty in the solution starting from partially unknown parameters.

The authors refer to the neural network approximating the spectral coefficients as a PINN however it seems that the residual of the convection-diffusion PDE is not considered in the loss function (4), hence this point appears to be somehow misleading.

This contribution shows a weakness point, related to the numerical results: only a 1D test case is considered, involving a single stochastic parameters. I suggest further investigations and analysis on the accuracy/efficiency of the proposed technique, at least on 2D problems, with larger stochastic dimension.

TYPOS

Line 62: hyperpolic

Line 94: as displayed and displayed

Line 154: the approximation of ??

**Score:**

2: Borderline paper

---

### Decision · Program_Chairs · 2021-10-17

**Decision:**

Accept (Poster)

**Comment:**

This paper considers extending Physics Informed Neural Networks (PINNs) to stochastic PDEs and thereby be able to quantify uncertainties in their predictions. The authors may consider this opportunity to extend the experimental section to address limitations of the work that has been flagged by reviewers.